# List-wise learning to rank biomedical question-answer pairs with deep ranking recursive autoencoders

**Yan Yan**[1]*, **Bo-Wen Zhang**[2], **Xu-Feng Li**[1], **Zhenhan Liu**[1]

**1** Department of Computer Science and Technology, School of Mechanical Electronic and Information Engineering, China University of Mining and Technology Beijing, Beijing, China, **2** Alibaba Group, Hangzhou, China

* yanyan@cumtb.edu.cn

**Data Availability Statement:** All relevant data are within the paper and its Supporting information files.

**Funding:** The funder, Aibaba Group, provided support in the form of salaries for authors [BZ], but

## Abstract

Biomedical question answering (QA) represents a growing concern among industry and academia due to the crucial impact of biomedical information. When mapping and ranking candidate snippet answers within relevant literature, current QA systems typically refer to information retrieval (IR) techniques: specifically, query processing approaches and ranking models. However, these IR-based approaches are insufficient to consider both syntactic and semantic relatedness and thus cannot formulate accurate natural language answers. Recently, deep learning approaches have become well-known for learning optimal semantic feature representations in natural language processing tasks. In this paper, we present a deep ranking recursive autoencoders (rankingRAE) architecture for ranking question-candidate snippet answer pairs (Q-S) to obtain the most relevant candidate answers for biomedical questions extracted from the potentially relevant documents. In particular, we convert the task of ranking candidate answers to several simultaneous binary classification tasks for determining whether a question and a candidate answer are relevant. The compositional words and their random initialized vectors of concatenated Q-S pairs are fed into recursive autoencoders to learn the optimal semantic representations in an unsupervised way, and their semantic relatedness is classified through supervised learning. Unlike several existing methods to directly choose the top-K candidates with highest probabilities, we take the influence of different ranking results into consideration. Consequently, we define a listwise "ranking error" for loss function computation to penalize inappropriate answer ranking for each question and to eliminate their influence. The proposed architecture is evaluated with respect to the BioASQ 2013-2018 Six-year Biomedical Question Answering benchmarks. Compared with classical IR models, other deep representation models, as well as some state-of-the-art systems for these tasks, the experimental results demonstrate the robustness and effectiveness of rankingRAE.

did not have any additional role in the study design, data collection and analysis, decision to publish, or preparation of the manuscript. The specific roles of these authors are articulated in the 'author contributions' section, and we don't have any funding.

**Competing interests:** We have the following interests: (BZ) is employed by Alibaba Group. There are no patents, products in development or marketed products to declare. This does not alter our adherence to all the PLOS ONE policies on sharing data and materials.

## Introduction

Due to the continuous growth of information produced in the biomedical domain, there is a substantially growing demand for biomedical QA from the general public, medical students, health care professionals and biomedical researchers [1]. Public demand for biomedical knowledge or access to natural knowledge is on the rise, especially regarding prevention methods and disease symptoms: medical students find relevant knowledge in papers or from work, while researchers follow the research results from previous studies. Moreover, biomedical QA is the most significant component of several real-world medical applications [2].

In recent years, various methods have been proposed in the field of biomedical QA [3]. It is known from experience that the current typical QA models or systems consist of three main parts: question processing, document processing, and answer processing phases [4, 5]. The question processing phase is usually responsible for converting questions in natural language expressions into queries which are suitable for a document search engine. Afterwards, the document processing phase controls retrieval of the most relevant documents with the generated queries and extracts candidate answer passages. Finally, in the answer processing phase, the candidate answers are matched against the expected answer type and are ranked according to the matching scores [6–8].

There have been several investigations concerning improvements of the query processing phase. For example, Cao et al. [9], Wasim et al. [10] and Abacha et al. [11] have employed question classifying approaches, with semantic information obtained from the UMLS resources. However, some researchers have noted that these medical QA approaches have limitations in terms of the types and formats of questions that they can process [12]. In contrast to the above studies focusing on query processing, several systems have been developed [13–15] with different document processing approaches. Standard IR [16] engines, e.g., Google, biomedical query systems, e.g., PubMed, or their combination have been proposed to return relevant documents in response to a query. In addition, some researchers have managed to utilize semantic knowledge in document retrieval [17, 18]. However, the statistics indicate that passage extraction can benefit more from incorporation of semantics as compared to document retrieval.

As a consequence, besides appropriate question analysis and document retrieval process, effectively extracting and selecting relevant answers represents the bottleneck in the entire process. From our own perspective, investigating how to select relevant snippets (explained in detail in Remark 1) directly from retrieved documents is significant in overcoming the limitations of question type and improving the performances to a large extent.

There is not much research on returning relevant snippets for biomedical questions. A previous study utilizing NCBI [19] suggested that the cosine similarities between questions and sentences in relevant documents represent the question-answer relationships. The study argued that the higher similarities represent higher relevances. There are two studies that ignore the differences between QA and IR. One is a study of BioASQ participants that builds a model with a granularity of several random words and calculates a ranking of the subdocument level through a document retrieval model [20]. Another one is a study of BioNLP participants utilizing encoder technology to measure the relationship between questions and answers [21]. Despite the improvements of performance, the above extracting strategies may break the completeness of semantics, whether with respect to the use of "sentence" or the definition of "granularity". Actually, in most cases, it is possible for a relevant snippet to be a single sentence or multiple sequential sentences, or even half a sentence, and the study of NCBI and BioNLP participants both exhibit inadequacy in that regard. According to our experiences, the snippets with the most similar keywords or term distributions are probably not the requested

answers. For instance, if the relevant documents happen to contain the exact statement of the question, then the expected answer is obviously the following sentences, rather than the similar sentence.

In this paper, we suppose that there are some latent semantic relations between a biomedical question and its relevant snippet answers, namely, a Q-A relation. Thus, the problem of selecting relevant snippet answers can be converted into several classification tasks to decide whether a question and the candidate answers have the Q-A relation. First, all possible candidate snippets are extracted from the documents, and each candidate snippet is combined with the question to form a question-snippet (Q-S) pair. Then, an appropriate vector representation model is utilized to represent the semantics of the Q-S pairs. Convolutional Neural Networks and Recurrent Neural Networks are both common vector representation models used to represent the global semantics, while the local semantics and the syntactic information may be ignored for comprehension. In contrast, Recursive Neural Networks (RNNs) maintain the local semantics to the utmost and take both syntactic and semantic information into account. As a result, RNNs are chosen to learn the semantic representations. Unlike the conventional classification, a specific prediction of Q-A relations may have various ranking results. Taking that fact into account, we modified RNNs by defining the "ranking error" and integrated it into loss function computation to correct the errors caused by ranking. With the semantic vectors of Q-S pairs and supervised learning, the probabilities of Q-A relations are computed and ranked to select relevant snippet answers.

We performed the experimental evaluations on the BioASQ 2013-2018 benchmarks with the Medline corpus. The results show that our proposed approach outperforms several competitive baselines, including the classical IR models and the proposed model with replaced vector representations, e.g., CNNs, LSTM and state-of-the-art BioASQ participants.

In summary, the main contributions are: 1) proposing a novel approach to solve the snippet retrieval problem in biomedical QA with a classification model; 2) redesigning the loss function of RNNs to orient ranking; and 3) providing a better solution for BioASQ.

**Remark 1** *"Snippet" is not an unambiguous concept like "sentence" or "paragraph". The exact definition is "a small and sequential piece of articles which represents an independent and complete semantic". The separators might be commas, dots, semicolons, or even the word "and". For instance, it could be a single sentence or a half sentence like "Most cases of CMT are caused by mutations in PMP22," or multiple sequential sentences like "PMP22 is the common gene found mutated through a duplication in CMT1A. Other genes are MPZ and SH3TC2.". So the extraction of snippets is a great challenge of snippets retrieval.*

## Related work

In a review several years ago, a few studies were highlighted that were dedicated to studying the biomedical question answering (QA) system [17, 22]. According to our research, Cairns et al. represented the first group to emphasize the importance of establishing a biomedical domain-specific question answering system. Then, TREC, one of the authoritative forums in the field of information retrieval based on large test collections related to QA systems, started a genomics track. Further, EQueR-EVALDA [23], a French evaluation campaign for question answering (QA) systems, provided two tasks: one of which is a biomedical domain-specific task to solve medical questions. Recently, there has been a huge range of success. In the sixth edition of the BioASQ challenge, 26 teams with more than 90 systems participated in this challenge in total, and the best ones were able to outperform the strong baselines [24]. Similarly to participants of the BioASQ challenge, participants of the BioNLP challenge also demonstrate very good performance [25]. However, there are still some limitations of biomedical QA, such

as lack of annotated data, ambiguity in clinical text and lack of comprehension of question/answer text by models [1, 26].

Apart from the tracks, organizations such as Google, MedQA [27], Onelook, and PubMed are also trying to construct question answering applications. With regard to the aspects of the quality of answers and ease of use, Google performs very well and better than the other three organizations [28]. All of them can return an acceptable response to the greater part of definitional questions posed by physicians. Due to some restrictions, however, only definitional questions can be solved. Another research project focused on retrieval of answers from biomedical literature through narrowing down the candidate answer space by question classification and distributing a higher rank to the correct answers [10]. This research still suffered from some troublesome problems [7, 29], such as the need for a clear factoid and list type.

In 2013, the first BioASQ challenge was held. Organizers of this challenge provided a large-scale question answering competition, in which the systems are required to cope with all stages of a question answering task, including the retrieval of relevant articles and snippets as well as the provision of natural language answers [30, 31]. The two teams, Choi S et al. [32] and Papanikolaou Y et al. [33], in this challenge proposed a model with a reference value. The third edition of the BioASQ challenge was hold in 2015. Sarrouti and El Alaoui [34] proposed using stemmed words and UMLS concepts as features for the BM25 model, which achieved good performances. The reason why they achieves good performances mainly because that they made full advantage of UMLS concepts and sentence components in both the document retrieval phase and the snippet retrieval phase. Their paper also proves that using the language resources of the sentence itself was equally important as using the model. A recent article also illustrates this fact [35]. The sixth edition of the BioASQ challenge was held in 2018 [24]. A team in this challenge took advantage of the theory of attention [36]. They used point multiplication of the query terms matrix and document terms matrix, like attention via dot-product, for encoding. They use pretrained embeddings with one dense layer and residual to generate context sensitive term encoding. Intuitively and rigorously, the context sensitive term encoding achieved the same effect with context encoding via the bidirectional RNN [37], and the former was faster. As a result, the system scored at the top or near the top for all tasks of this challenge [8]. The above models or systems have some defects: only some matching of relevant documents achieved successful results, according to the evaluation. When searching for relevant snippets, the results became terrible because the systems could not find the accurate positions of the relevant snippets [17]. However, as mentioned in the introduction, relevant documents cannot meet the requirements because the accurate statements are difficult to locate manually when given the candidate literature. Instead, relevant snippets can solve this issue. According to the overview of BioASQ competitions, most participants working on snippet retrieval adopted similar proposals to the methods while searching articles. The main differences were the methods used to split the documents. NCBI suggested the use of sentences directly in relevant documents [19]. Another study of BioASQ participants aimed to define a granularity of several words to split the documents [20]. There were also several researchers who regarded all possible snippets as different "short documents." The indices of these candidates were then built for preprocessing, and the same retrieval models were utilized to rank them. Apart from the retrieval approaches, the framework proposed by NCBI [38] directly computes the cosine similarities between the questions and the candidate sentences to measure their relatedness. Finally, the best scoring sentences from the title or the abstract were chosen as relevant snippets for a question.

From our perspective, these approaches excessively rely on the information retrieval techniques in which the ranking is based on the distributions of query terms in documents and the whole collection. There is severe weakness existing in these approaches due to lack of

consideration of the semantics. The cosine similarity represents the degree of resemblance rather than the Q-A relations. In a similar way, the output scores from any classical IR models also represent the similarities of term distributions in the questions/queries, in the documents, or in the whole collection. The semantic meanings are not taken into account when deciding whether they have Q-A relations, while the semantics are usually the definitive factors. For instance, for a biomedical question such as "How to treat infectious mononucleosis," a statement inside a candidate document is "What is the treatment for infectious mononucleosis? Chloroquine and steroids are worth attempting." Obviously, the expected relevant snippet is the latter sentence, "Chloroquine and steroids are worth attempting," rather than the former "What is the treatment for infectious mononucleosis?" Consequently, including semantics is of great importance to locate the relevant snippets for biomedical questions.

## Ranking with modified RNNs

As described above, the modified RNNs is used to generate a variable-size vector representing the Q-S pair to discover the semantic relations between the question and the candidate snippet. In this section, we respectively introduce the preprocessing work, the unsupervised RNNs, which recursively combine word vectors, and the modified semi-supervised RNNs, which both learn the semantic representations and solve the ranking problem. The architecture of modified RNNs, which learn the semantic vector representations of Q-S pairs and classify whether the Q-S pairs have Q-A relations, is shown in Fig 1.

### Preprocessing and pretraining

We first perform query formulation on the input questions and feed the generated queries into a search engine to retrieve relevant documents. Specially, in that step of query formulation, we use NLTK to create a parse tree of parts of speech on every input question and remove all non-noun phrase (NNP) parts, for, it is not enough to remove stop words. A typical problem is that

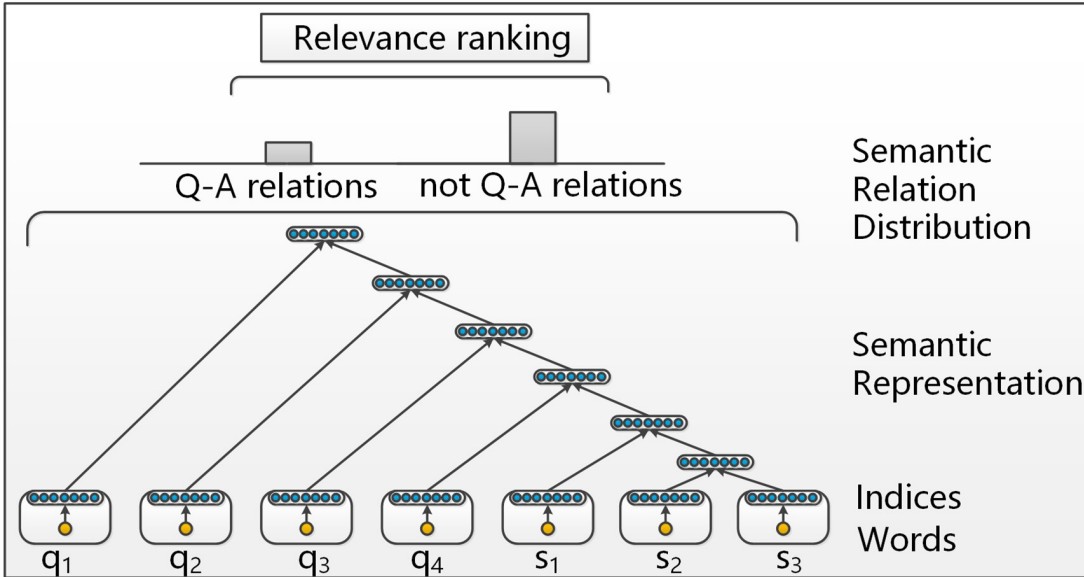

**Fig 1. Illustration of the modified RNNs architecture to learn semantic vector representations for a Q-S pair.** Words are pretrained first into continuous vectors. Then, they are recursively combined into a fixed length vector through the same autoencoders. The vectors at each node are used as features to predict the local semantic relations.

we cannot retrieve the documents that we need if we remove stop words from questions only. Through word frequency analysis, we find that most documents containing answers contain nouns or other forms of nouns in the question but other parts of speech do not appear regularly. So if we don't delete those words, we may not be able to retrieve all the documents we need since search engines tend to retrieve documents with more input. Experiments have found that leaving noun phrases works better than leaving nouns only. Then, all possible candidate snippets are extracted from top-N documents to guarantee the recall of ideal snippet answers. Each snippet and question are combined together into a Q-S pair.

Moreover, the semantic vectors of words are required. Random continuous vectors are usually used, but here, a coarse learning process is applied to pretrain the word vectors with the word2vec tool on the Medline article collection. With pretraining, the recursive iterations and the corpus impact can be effectively decreased.

## Recursive autoencoders and variants

The goal of autoencoders is to combine a sequence of word vectors into a single vector of fixed dimensions and size. At each step, it encodes two adjacent vectors that meet certain standard as a vector. For example we have a sequence $x = (x_1, x_2, x_3, x_4, x_5)$ and $(x_2, x_3)$ that meets the standard. It will be required to encode $(x_2, x_3)$ as a vector $y_1$. Then, a new sequence will be generated $x = (x_1, y_1, x_4, x_5)$ and it becomes shorter. We call $y_1$ the father node of $(x_2, x_3)$. After a few steps, the sequence will be encoded as a single vector and the track of encoder is a tree structure. Fig 2 shows an instance of recursive autoencoders (RAE) with a list of word vectors $x = (x_1, \ldots, x_m)$ and a binary tree structure. We chose a binary tree instead of a parse tree. This is because that the parse tree is built according to certain standards, and it can only encode vectors according to a fixed pattern. For binary tree, the pattern is selected by the neural network itself. It can constantly encode moreproperly and enable vectors to encode together. The tree structure can also be described with several triplets $p \rightarrow c_1 c_2$ where $p$ is the parent node and $c_1$, $c_2$ are the children, such as$(y_1 \rightarrow x_3 x_4, y_2 \rightarrow x_2 y_1, y_3 \rightarrow x_1 y_2)$. With the same neural networks, the parent representations p can be computed from the children $c_1, c_2$ with:

$$p = f(W^{(1)}[c_1 : c_2] + b^{(1)}) \tag{1}$$

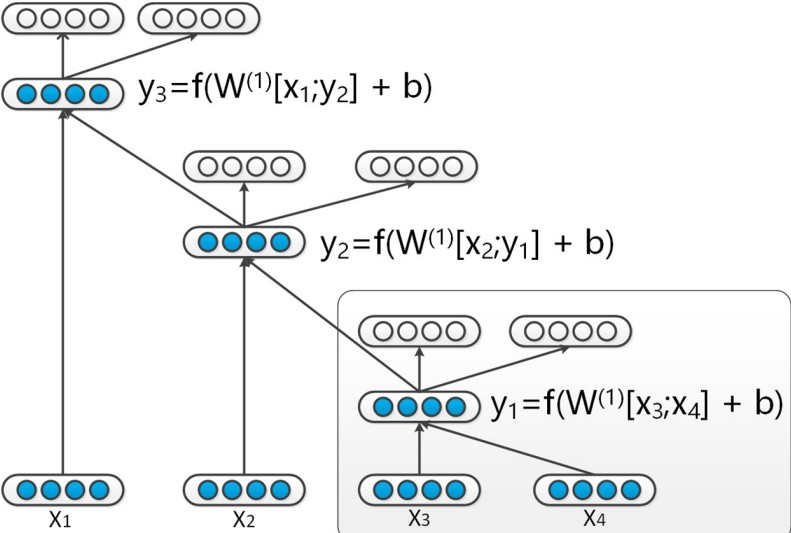

**Fig 2. Illustration of an application of a recursive autoencoder to a binary tree.** The white nodes are utilized to calculate the reconstruction errors.

where the concatenation of the two children is multiplied by a matrix of parameters $W^{(1)} \in \mathbb{R}^{n \times 2n}$. After adding a bias term $b$, the *tanh* is applied as activation function. A reconstruction layer is usually designed to validate the combination process by reconstructing the children with:

$$[c'_1 : c'_2] = W^{(2)}p + b^{(2)} \tag{2}$$

Then, through comparisons between the reconstructed and the original children vectors, the reconstruction errors can be computed by their Euclidean distance, as shown in:

$$E_{rec}([c_1; c_2]) = \frac{1}{2}\|[c_1; c_2] - [c'_1; c'_2]\|^2 \tag{3}$$

Now that the vector representation for a parent node $p$ of two children $(c_1, c_2)$ can be computed and the dimensions are the same, the full tree is constructed with the triplets and recursive combinations; as such, the reconstruction error at each nonterminal node is available. However, during the recursive process, the child node could represent a different number of words and, thus, different importance for the overall meaning reconstruction. We therefore adopt the strategy [39] to redefine reconstruction error as:

$$E_{rec}([c_1; c_2]; \theta) = \frac{n_1\|c_1 - c'_1\|^2 + n_2\|c_2 - c'_2\|^2}{n_1 + n_2} \tag{4}$$

where the $n_1$ and $n_2$ represent the number of words in $(c_1, c_2)$ and $\theta$ stands for the parameters.

To minimize the reconstruction errors of all vector pairs of children in a tree, the tree structure can be computed through:

$$RAE_\theta(x) = \arg\min_{y \in A(x)} \sum_y E_{rec}([c_1; c_2]; \theta) \tag{5}$$

where $A(x)$ stands for the set of all possible trees that can be built from an input Q-S pair $x$. According to [39], a greedy approximation can simplify the tree construction. For each time, the "potential" parent node and reconstruction error of each pair of neighboring vectors are calculated, and the pair with the lowest reconstruction error is replaced by a parent node. This process is repeated until the sequence is encoded as a single vector, and a encoding tree is also constructed completely. This approximation captures single-word information to a large extent and does not necessarily follow syntactic constraints; it even breaks the boundaries between questions and snippet, which may help to decide whether a question and a snippet are naturally connected and also solve the problem that the length of the sentence is not equal or too long.

## Semi-supervised modified RNNs for ranking

With unsupervised RAE, the semantic vectors of Q-S pairs are generated. We extend the approach into a semi-supervised RNNs to predict the semantic relations and rank the potentially relevant snippets for a question. The distributed vector representation of each parent node in the tree built by RAE could also be regarded as features of the Q-S pairs, so we leverage the vector representations by adding a simple softmax layer on top of each parent node to predict class distributions. This is a multi-task learning structure, with encoder as the main task and classification as the branch task. The classification layer will affect the encoding results, making the encoder to generate vectors that are more friendly and suitable for classification,

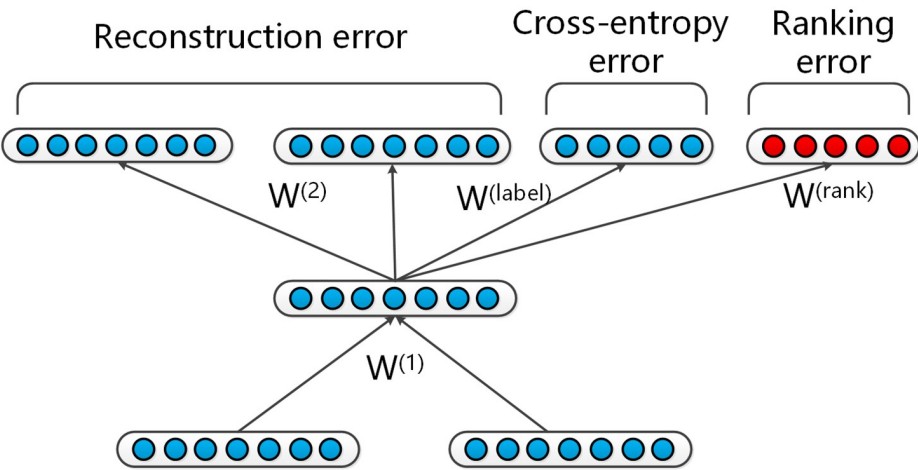

**Fig 3. Illustration of a unit in modified RNNs at a nonterminal node.** The red nodes shows the ranking error.

and therefore achieve the purpose of improving accuracy:

$$d(p; \theta) = softmax(W^{label}p) \tag{6}$$

Fig 3 shows a unit in the modified RNNs at a parent node. Let $d = (d_1, d_2)$, $d_1 + d_2 = 1$ represent the distribution with and without Q-A relations, and $t_1$, $t_2$ be the target label distribution for one entry. Since the outputs of the *softmax* layer are conditional probabilities $d_k = p(k|[c_1; c_2])$, the cross-entropy error can be computed with:

$$E_{cE}(p, t; \theta) = -\sum_{k=1}^{2} t_k \log d_k(p; \theta) \tag{7}$$

So the training error for each entry can thus be computed through the sum over the error at the nodes of the tree $T$:

$$E(x, t; \theta) = \sum_{s \in T} E([c_1; c_2]_s, p_s, t, \theta) \tag{8}$$

where the error at each nonterminal node is the weighted sum of reconstruction and cross-entropy errors:

$$\alpha E_{rec}([c_1; c_2]_s; \theta) + (1 - \alpha) E_{cE}(p_s, t; \theta) \tag{9}$$

As mentioned, the modified RNNs are in charge of not only classifying the Q-S pairs but also ranking the candidate snippet answers according to the value of relevance. However, we have found that the same classification result may lead to different ranking results due to the influence among samples, which cannot be measured with cross-entropy error. For example, a question $q$ has a relevant answer $s_1$ and an irrelevant answer $s_2$. The target label distributions of $qs_1$, $qs_2$ are thus $(1, 0)$, $(0, 1)$ respectively. Assume that there are two classifiers with predictions $(0.51, 0.49)$, $(0.52, 0.48)$ and $(1, 0)$, $(0.99, 0)$. With classification, the two classifiers have the same results, where candidate snippets $s_1$, $s_2$ are both relevant. The cross-entropy errors of the latter classifier are much larger than those of the former one. However, if the top-1 answer is requested, the latter would make the correct selection. In fact, the ranking accuracy is much more significant than classification accuracy in this case.

The above instance indicates that the training error of each entry is influenced by estimated probabilities of the other entries, which corresponds to the same question. Hence, we define the "ranking error" to represent the training error associated with the ranking process.

Assume that there is a set of top $N$ candidate snippets $C = \{x^{(1)}, x^{(2)}, \ldots, x^{(N)}\}$ for a biomedical question and a set of representation vectors for Q-S pairs $P = \{p^{(1)}, p^{(2)}, \ldots, p^{(N)}\}$. Let $D = \{d^{(1)}, d^{(2)}, \ldots, d^{(N)}\}$ be the set of output distributions, where $d^{(i)} = (d_1^{(i)}, d_2^{(i)})$. To avoid confusion, we assume that $x^{(1)}, x^{(2)}, \ldots, x^{(m)}$ are relevant and the rest are irrelevant. The set of target label distributions is therefore $L = \{t^{(1)}, t^{(2)}, \ldots, t^{(N)}\}$, $t^{(i)} = (1, 0)$, $i \leq m$ and $t^{(i)} = (0, 1)$, $i > m$. According to the values of $d_1^{(i)}$, the rank $r$ of the candidate snippets $C$ can be computed through $r = rank(D) = rank(d(P;\theta))$. In addition, $m$ equals the number of $t^{(i)} = (1, 0)$, that is $m = count(L)$.

Mean Average Precision (MAP) is a global evaluation metric to measure the ranking results, so the ranking error is defined as the negative of the logarithm of the MAP score:

$$E_r(P, L; \theta) = -\log\left(\sum_{i=1}^{m} \frac{d_1^{(i)}}{i}\right) = -\log\left(\sum_{i=1}^{count(L)} \frac{d(P; \theta)_1^{(i)}}{i}\right) \tag{10}$$

As a result, the loss function which corresponds to a question $E'(C, L;\theta)$ can be computed by following Eq (11), while the final objective and its gradient are respectively shown in Eqs (12) and (13).

$$\beta E_r(P, L; \theta) + (1 - \beta) \sum_{t \in L x \in C} E'(x, t; \theta) \tag{11}$$

$$J = \frac{1}{N} \sum_{(x,t)} E'(C, L; \theta) + \frac{\lambda}{2} \|\theta\|^2 \tag{12}$$

$$\frac{\partial J}{\partial \theta} = \frac{1}{N} \sum_{(C,L)} \frac{\partial E'(C, L; \theta)}{\partial \theta} + \lambda \theta \tag{13}$$

With proper learning through the modified RNNs, the probability of Q-A relations within a Q-S pair can be estimated by the output distributions. The candidate snippet answers are then ranked according to the estimated probabilities of the corresponding Q-S pairs, and the top ranked snippets are predicted to be relevant.

## Experiments

### Experimental evaluation

We evaluate the performance of our method on the biomedical literature collection from PubMed/MedLine [40] and the benchmark datasets of questions from the three-year BioASQ challenges [20, 24]. The literature collection contains a total of over 20 million records which contain the article title and abstract. The benchmark datasets contain several questions that are requested to reflect real-life information needs encountered during the work, research or diagnosis of several biomedical professionals. Moreover, each question should be independent, i.e., it should not contain any pronouns referring to entities mentioned in other questions. The ground truth of each question and the supportive information are also provided by these experts. The questions are categorized into four classes [41, 42]: (1) yes/no questions, (2) factoid questions, (3) listed questions, which respectively require a "yes"/"no" answer, a particular entity (e.g., a disease, drug, or gene), or a list of entities as an answer, and (4) Summary

questions that can only be answered by producing short text summarizing the most prominent relevant information; e.g., "*What is the treatment for infectious mononucleosis?*".

There are respectively 3, 5, and 5 batches in the three-year BioASQ challenge, each batch containing 100 questions. For snippet retrieval, participants are asked to submit at most 10 relevant snippets extracted from the literature.

**Algorithm comparison.** We compare the performance of the proposed method with several strong baselines. Specifically, in order to validate the effectiveness of components, we implement some baselines by replacing the vector representation in our model with several sentence models, including *Convolutional Neural Networks (CNN)* [43, 44], *Recurrent Neural Networks (RNN)* [45, 46], *Long Short Term Memory (LSTM)* [47], and the original *RAE*, to validate the necessity of the ranking error definition. Some classical IR models provided by an open-source search engine are also chosen as baselines to verify the use of classification, including *Query Likelihood (QL)*, *Sequential Dependence Model (SDM)* and *BM25* [48]. Moreover, the participating systems developed by the challenge winners of the six-year BioASQ are also baselines.

For all experiments, the sets of candidate documents are retrieved based on a unified index construction and IR model provided by an open-source search engine, Galago http://www.lemurproject.org/galago.php, with default settings.

## Results

### Comparisons with variants of our approach

We present the performances of our proposed approach https://github.com/lixuf/RAE-Recursive-AutoEncoder-for-bioasq-taskB-phaseA-snippets-retrieve- and the variants that replace the vector representation model with self-implemented Convolutional Neural Networks (CNN), Recurrent Neural Networks (RNNs), Long Short-term Memory (LSTM), and the RAE without the use of ranking error (RAE). Evaluated with the BioASQ official metric of Mean Average Precision (MAP), the comparison results with these variants are reported in Table 1. The results show that our approach performs better than all of the variants across all datasets. Specifically, if we use the names of representation models to stand for the baselines, then in terms of BioASQ 2013, our method outperforms CNN, RNN, LSTM, and RAE by 36.2%, 30.0%, 26.8% and 18.6% over the 3 batches; with respect to BioASQ 2014, the CNN,

**Table 1. The MAP performances of our approach compared with the variants and classical IR models on BioASQ.**

| Dataset | Batch | Our | CNN | RNN | LSTM | RAE | QL | SDM | BM25 |
|---|---|---|---|---|---|---|---|---|---|
| BioASQ 2013 | Batch 1 | **0.0822** | 0.0642 | 0.0675 | 0.0694 | 0.0736 | 0.0564 | 0.0583 | 0.0546 |
| | Batch 2 | **0.0631** | 0.0450 | 0.0486 | 0.0497 | 0.0523 | 0.0354 | 0.0372 | 0.036 |
| | Batch 3 | **0.0795** | 0.0559 | 0.0568 | 0.0582 | 0.0637 | 0.0536 | 0.0548 | 0.0527 |
| BioASQ 2014 | Batch 1 | **0.0892** | 0.0524 | 0.0568 | 0.0571 | 0.0783 | 0.0586 | 0.0650 | 0.0524 |
| | Batch 2 | **0.0656** | 0.0478 | 0.0493 | 0.0506 | 0.0612 | 0.0465 | 0.0478 | 0.045 |
| | Batch 3 | **0.0795** | 0.0465 | 0.0483 | 0.0498 | 0.0624 | 0.0542 | 0.0563 | 0.0517 |
| | Batch 4 | **0.0743** | 0.0482 | 0.0490 | 0.0503 | 0.0617 | 0.0510 | 0.0536 | 0.0493 |
| | Batch 5 | **0.0668** | 0.0482 | 0.0476 | 0.0485 | 0.0523 | 0.0518 | 0.0523 | 0.0529 |
| BioASQ 2015 | Batch 1 | **0.0724** | 0.0374 | 0.0397 | 0.0416 | 0.0539 | 0.0386 | 0.0429 | 0.0378 |
| | Batch 2 | **0.0931** | 0.0589 | 0.0603 | 0.0641 | 0.0685 | 0.0594 | 0.0648 | 0.0592 |
| | Batch 3 | **0.1048** | 0.0762 | 0.0824 | 0.0863 | 0.0932 | 0.0856 | 0.0895 | 0.0873 |
| | Batch 4 | **0.1056** | 0.0945 | 0.0938 | 0.0976 | 0.0960 | 0.0895 | 0.0928 | 0.0864 |
| | Batch 5 | **0.1412** | 0.1190 | 0.1052 | 0.1131 | 0.1203 | 0.1178 | 0.1201 | 0.1165 |

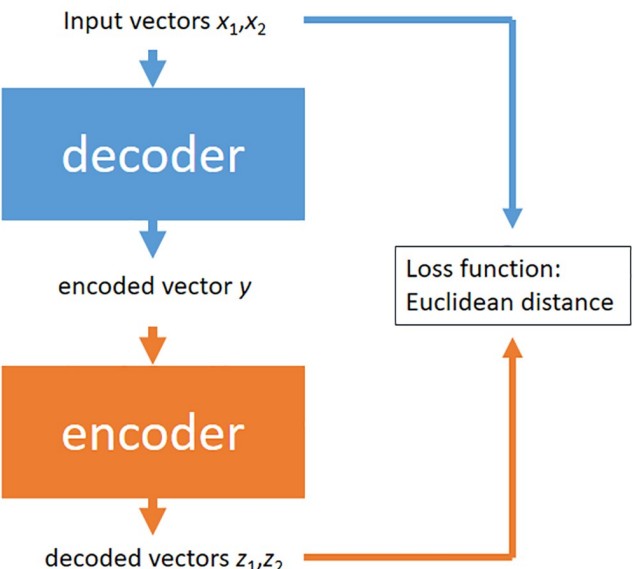

**Fig 4. The input vectors $x_1$, $x_2$ are the children mentioned in recursive autoencoders and variants section, and the encoded vector $y$ is the parent mentioned in recursive autoencoders and variants section.**

RNN, LSTM, and RAE are improved by 59.4%, 49.6%, 46.5% and 18.9% on average; and in terms of BioASQ 2015, the average improvements of performance are 34.0%, 35.6%, 28.4% and 19.7%, respectively. The loss function of RAE is the Euclidean distance between the input vectors and the decoded vectors, is shown in Fig 4. Therefore, the goal of each iteration of RAE is to obtain the decoded vectors with the highest similarity to the input vectors, so as to obtain the hidden layer state (encoded vector) that can be put into the decoder and get the vectors with a high degree of similarity to the input vectors. We can visually think of the encoder as a compression tool, which compresses the input vectors into a vector. It is worth noting that the structure of the encoder and the decoder must be consistent while the data flow should be opposite. This is why we can use the similarity between the decoded vectors and the input vectors as a criterion. It was found that the input vectors $x_1$, $x_2$ encoder to vector $y$ after a series of compute and the vector $y$ decoder to the vectors that are highly similar to the input vectors $x_1$, $x_2$ again after a series of compute. The reason why the vector $y$ can be returned to the input vectors is that the vector $y$ has most of the features of the input vectors similar to word embedding. So RAE can retain as much of the local semantics as possible, which is its goal also.

Obviously, from the statistics and the analysis above, the vector representation model in our proposed approach is more suitable than other vector representation models to a large extent on this task. Among these variants, RAE is substantially better than others. Moreover, LSTM is slightly better than CNN or RNN, except for some individual batches. From our perspective, the CNN aims to discover the full depth of the input sentences with a global pooling operation, which is appropriate for learning the global semantics, while the RNN or LSTM generate the semantic vectors with the sequential models, which are usually utilized to predict the next words in a sequence. However, for the Q-S pairs, we are more concerned about the relations rather than the precise semantics, which is supported by the statistics of CNN. Additionally, from the comparisons of CNN and RNN/LSTM, we have found that the sequentiality is beneficial to the judgments of the Q-A relations to a certain degree but still does not represent the decisive factor. Moreover, the results of RAE prove the significance of maintaining the local semantics for accurate judgments. In addition, the comparisons with RAE manifest

our novel design that takes the "ranking error" into consideration during the loss function computation.

## Comparisons with classical IR models

As mentioned above, the entire process of answer matching and ranking can be regarded as snippet retrieval, so we also compare our approach with classical IR models, including the query likelihood model (QL), BM25 and sequential dependence model (SDM). The exact MAP scores on BioASQ 2013-2015 are also shown in Tables 1 and 2–7 show the MAP scores

**Table 2. Comparisons with BioASQ 2013 participants.**

| System | Batch 1 | Batch 2 | Batch 3 |
|---|---|---|---|
| our | **0.0822** | **0.0631** | **0.0795** |
| Wishart | - | 0.0360 | - |
| BAS 100 | 0.0578 | 0.0337 | 0.0537 |
| BAS 50 | 0.0512 | 0.0272 | 0.0527 |

**Table 3. Comparisons with BioASQ 2014 participants.**

| System | Batch 1 | Batch 2 | Batch 3 | Batch 4 | Batch 5 |
|---|---|---|---|---|---|
| our | **0.0892** | **0.0656** | **0.0795** | **0.0743** | **0.0668** |
| Wishart | 0.0364 | 0.0379 | 0.0574 | 0.0503 | 0.0476 |
| main system | 0.0095 | 0.0062 | - | - | - |
| Biomedical Text Ming | 0.0296 | - | 0.0215 | 0.0240 | 0.0195 |
| BAS 100 | 0.0608 | 0.0319 | 0.0486 | 0.0549 | 0.0544 |
| BAS 50 | 0.0601 | 0.0313 | 0.0480 | 0.0539 | 0.0539 |
| HPI-S1 | - | 0.0482 | 0.0517 | 0.0300 | - |

**Table 4. Comparisons with BioASQ 2015 participants.**

| System | Batch 1 | Batch 2 | Batch 3 | Batch 4 | Batch 5 |
|---|---|---|---|---|---|
| our | 0.0724 | 0.0931 | 0.1048 | 0.1056 | 0.1412 |
| ustb_prir | 0.0797 | 0.0776 | 0.1840 | 0.2005 | 0.2410 |
| qaiiit | 0.0789 | **0.1159** | - | 0.1415 | - |
| HPI | **0.0971** | 0.0719 | 0.1269 | 0.1627 | 0.1341 |
| testtext | 0.0752 | 0.0817 | 0.1128 | 0.2070 | - |
| oaqa | - | - | **0.1969** | 0.2092 | 0.2196 |
| fdu | - | - | 0.1166 | **0.2480** | **0.2424** |

**Table 5. Comparisons with BioASQ 2016 participants.**

| System | Batch 1 | Batch 2 | Batch 3 | Batch 4 | Batch 5 |
|---|---|---|---|---|---|
| our | 0.1247 | 0.1392 | 0.1701 | 0.2300 | 0.2401 |
| KNU-SG Team_Korea | 0.1365 | 0.1590 | 0.1693 | 0.2305 | 0.2386 |
| ustb_prir | 0.0700 | 0.0884 | 0.1127 | 0.2298 | 0.2250 |
| testtext | 0.0641 | 0.0774 | 0.1069 | 0.1834 | 0.1694 |
| fdu | - | **0.1870** | **0.2214** | **0.2365** | **0.2882** |
| HPI | **0.1601** | - | 0.1696 | - | 0.2049 |

**Table 6. Comparisons with BioASQ 2017 participants.**

| System | Batch 1 | Batch 2 | Batch 3 | Batch 4 | Batch 5 |
|---|---|---|---|---|---|
| our | 0.1402 | 0.1598 | 0.1524 | 0.1726 | 0.1847 |
| KNU-SG Team_Korea | 0.1393 | 0.1734 | 0.1411 | 0.1385 | - |
| ustb_prir | **0.1747** | **0.2598** | **0.2727** | **0.2423** | **0.2090** |
| testtext | 0.1585 | 0.2523 | 0.3500 | 0.2465 | 0.1843 |
| fdu | - | 0.1711 | 0.3183 | 0.1436 | 0.1170 |

**Table 7. Comparisons with BioASQ 2018 participants.**

| System | Batch 1 | Batch 2 | Batch 3 | Batch 4 | Batch 5 |
|---|---|---|---|---|---|
| our | 0.1189 | 0.1628 | 0.1950 | 0.1102 | 0.0895 |
| MindLab | 0.0004 | 0.2736 | 0.2217 | 0.1413 | 0.1006 |
| ustb_prir | 0.1209 | 0.1731 | 0.2021 | 0.1216 | 0.0967 |
| testtext | 0.1151 | 0.1463 | 0.2021 | 0.1213 | 0.0861 |
| aueb | **0.1684** | **0.3187** | **0.3320** | **0.2138** | **0.1147** |

on BioASQ 2013-2018. The statistics in the table indicate that our approach extensively outperforms the QL, SDM and BM25 for all batches of three-year datasets. Compared to QL, the average improvements on BioASQ 13-15 are respectively 54.6%, 43.2% and 32.3%; our approach exhibits a great advantage over SDM by the average improvements of 49.6%, 36.5% and 26.1%; the average improvements are even larger with respect to BM25, which are respectively 56.9%, 49.4% and 33.5%.

Among these IR models, SDM performs better than QL and BM25. This improvement is mainly because SDM focuses on the sentence structure of queries and documents. The possible phrases in sentences are considered through the exact phrase feature and unordered window feature during retrieval, which is similar to the remaining local semantics in our approach. QL and BM25 are mainly based on the term distributions in queries and documents, which lack the consideration of semantics. Therefore, our proposed approach is more suitable than these IR models to retrieve the relevant snippets for biomedical questions.

Unlike the preceding year, quite a few teams participated in BioASQ 2014 [20], and most of the submitted results were well-performed. The performances of our approach and challenge winners are shown in Table 3. The *Wishart* team utilized a similar strategy in BioASQ 2013. The *NCBI* team's framework used the cosine similarity between question and sentence to compute their similarity. The *HPI* team relied on the Hana Database and BioPortal to retrieve biomedical concepts and merged the concepts to retrieve the snippets.

In BioASQ 2015, semantic vectors were first applied among the participants (*ustb_prir* team) [49] to look up the synonyms of the keywords in queries to select effective terms for query expansion. The *oaqa* team [50] proposed a collective reranking model with supervised learning. The *qaiiit* team [51] applied snippet extraction based on the similarity of the top 10 sentences of the retrieved documents and the queries. The evaluation results are demonstrated in Table 4.

In BioASQ 2016, *HPI-S1* [52] was based on the existing NLP functions from a in-memory database (IMDB) and it was extended with a new process specifically to QA. *KNU-SG* [53] proposed a system using a cluster–based language model. *WS4A* [54] proposed a novel approach consists on the maximum exploitation of existing web services. The evaluation results are shown in Table 5.

In BioASQ 2017, Brokos etc. [55] proposed a retrieval method that represents documents and questions as weighted centroids of word embeddings and reranks the retrieved documents with a relaxation of Word Mover's Distance. *USTB_PRIR* [56] introduced different multi-modal query processing strategies to enrich query terms and assign different weights to them. The evaluation results are shown in Table 6.

In BioASQ 2018, *MindLab* [26] proposed a model making use of semantic similarity patterns that were evaluated and measured by a convolutional neural network architecture. *AUEB* [8] used novel extensions to deep learning architectures. The evaluation results are shown in Table 7.

From the statistics in the tables, we can see that our approach improves the best participating systems in BioASQ 2013-2015 by 52.4%, 36.1% and 18.0%, respectively. In BioASQ 2016-2018, our model performed close to the best competitors and even prevailed in some batches. The decreases of improvements do not indicate the decline of robustness. The ultimate causes are the introduction of extra resources, for example, introducing a pre-trained document retrieval model at the stage of retrieving documents can not only retrieve more comprehensively, but also reduce the probability of retrieving useless documents greatly. Using the more primitive search tool provided by pubmed, the top 100 relevant documents contain an average of 4.3 target documents, which can barely be called a comprehensive search. However, introducing too many useless documents brings a large error to the classification model. If extra resources of a pre-trained document retrieval model are introduced, the compression ratio of useful documents to useless documents can be reduced to 1:2 or even lower during the retrieval phase. Especially after BioASQ 2015, most of the systems based on extra resources contain a large amount of domain knowledge in biomedicine. In addition, extra resources of language system, like UMLS, can help the model to better calculate the relationship between the problem and the paragraph through the connection between medical concepts or vocabulary. [34] The word frequency is used to represent the degree of professionalism of the vocabulary, q-s pairs containing vocabulary with a word frequency less than 15 can be selected and retrieved in UMLS, and then be put the selected q-s pairs encoded with our model into the classification model containing Attention and output the results of the Attention. After standardizing the result of Attention, each word's the degree of influence in the q-s pairs of the final classification result can be obtained, which is a decimal between 0 to 1. A larger value indicates that the word has a greater influence on the final result. It is found that more than half of these highly specialized vocabularies have a low impact on the final result. But these specialized vocabulary and many professional concepts associated with the vocabulary are the key to a correct answer. In summary, the proposed modified RNNs represent a practical approach to retrieve relevant snippets for biomedical questions compared with the state-of-the-art [57] BioASQ participants.

## Significance testing and experimental analysis

To report effect sizes and confidence intervals more informatively, we performed several two-sided paired t-test experiments between our approach and each self-implementing approach on all 13 batches, including the variants and the IR models, according to Tetsuya Sakai's significance testing [58]. According to the two-sided paired t-test experiments for the difference in means $\bar{d} = 0.0249$ (with the unbiased estimate of the population variance V = 0.0008), our approach statistically significantly outperforms *CNN* (t(13) = 3.1946, p < 0.0077, $ES_{pairedt}$ = 0.8860, 95% CI [0.0079,0.0418]). The exact results of the other comparisons are shown in Table 8. Obviously, we can observe that all p-values are less than **0.01**.

**Table 8. Two sided paired t-test results on our approach with the baselines.**

|  | $\bar{d}$ | $t_0$ | p(<) | ES | 95% CI |
|---|---|---|---|---|---|
| CNN | 0.0249 | 3.1946 | 0.0077 | 0.8860 | [0.0079, 0.0418] |
| RNN | 0.0240 | 3.1779 | 0.0080 | 0.8814 | [0.0075, 0.0405] |
| LSTM | 0.0216 | 3.1602 | 0.0082 | 0.8765 | [0.0067, 0.0365] |
| RAE | 0.0138 | 3.1337 | 0.0086 | 0.8691 | [0.0042, 0.0235] |
| QL | 0.0245 | 3.2535 | 0.0069 | 0.9024 | [0.0081, 0.0410] |
| SDM | 0.0217 | 3.2567 | 0.0069 | 0.9033 | [0.0072, 0.0362] |
| BM25 | 0.0258 | 3.2427 | 0.0071 | 0.8994 | [0.0085, 0.0431] |

Through the above comparisons and the statistical significance testing, we can conclude that our approach outperforms other vector representation models, IR models and state-of-the-art BioASQ participants. There are several reasons leading to the improvements. First, our approach aims to discover the semantic relations by classifying the Q-S pairs, while the purposes of classical IR models and some BioASQ participants are to measure the similarities of term distributions or semantics between the question and the candidate answers. Second, during the vector representation process, our approach retains as much of the local semantics as possible, which may benefit the classification of Q-S pairs. There is a typical model that get results by measuring the similarities of term distributions [53]. It is difficult to consider comprehensively, although this paper proposed six aspects to measure the similarity between questions and answers. The model proposed by this paper also requires a lot of expertise and experimentation to determine which aspects to be used. Our proposed model encodes Q-S pair automatically instead of the similarities of term distributions. Our approach not only needs less expertise and experiments but also automatically selects the required information by neural network and has better result than that model. Another paper [26] also uses the similarities of term distributions, but they propose a similarity matrix generated by part-of-speech and similarity. The form of matrix can increase their computing speed because they have a document retrieval model that can provide more accurate related documents than search engines, which is one of our weaknesses. Another disadvantage of our model is that we don't take full advantage of part of speech, position and similarity, but they do. Two methods provided us with improved ideas above. Combining our method with their method may achieve better results but requires more computing power and manpower.

## Conclusion

This paper studies the problem of answer matching and ranking issues for biomedical question answering with respect to a modified RNNs model. Our approach features the following novelties. (1) The proposed model successfully converts a snippet retrieval problem for biomedical questions into several classification tasks judging the semantic relations between biomedical questions and the candidate snippets. (2) The modified RNNs proposed a brand new definition—"ranking error"—in the loss function computation, which makes the conventional recursive neural networks more suitable for a ranking problem. (3) The proposed approach provides a simple but effective snippet retrieval proposal for the development of a biomedical question answering system. As relevant issues for future work, there are two directions. One direction is to extend our model to the semantic search of short text within the open domain. The other is to popularize the "ranking error" to make other classification models suitable for ranking.

## Supporting information

**S1 File.**
(RAR)

## Author Contributions

**Conceptualization:** Yan Yan, Bo-Wen Zhang, Zhenhan Liu.

**Data curation:** Yan Yan, Bo-Wen Zhang, Xu-Feng Li, Zhenhan Liu.

**Formal analysis:** Yan Yan, Bo-Wen Zhang, Xu-Feng Li, Zhenhan Liu.

**Funding acquisition:** Yan Yan, Bo-Wen Zhang, Zhenhan Liu.

**Investigation:** Yan Yan, Bo-Wen Zhang, Xu-Feng Li, Zhenhan Liu.

**Methodology:** Yan Yan, Bo-Wen Zhang.

**Project administration:** Yan Yan, Bo-Wen Zhang.

**Resources:** Yan Yan, Bo-Wen Zhang.

**Software:** Yan Yan, Bo-Wen Zhang, Xu-Feng Li.

**Supervision:** Yan Yan, Bo-Wen Zhang.

**Validation:** Yan Yan, Bo-Wen Zhang.

**Visualization:** Yan Yan, Bo-Wen Zhang.

**Writing – original draft:** Yan Yan, Bo-Wen Zhang.

**Writing – review & editing:** Yan Yan, Bo-Wen Zhang, Xu-Feng Li.

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
