## [Decision Letter · Decision Letter 0]

6 Jan 2020

PONE-D-19-28201

List-wise Learning to Rank Biomedical Question-Answer Pairs with Deep Ranking Recursive Autoencoders

PLOS ONE

Dear Miss. Yan,

Thank you for submitting your manuscript to PLOS ONE. After careful consideration, we feel that it has merit but does not fully meet PLOS ONE’s publication criteria as it currently stands. Therefore, we invite you to submit a revised version of the manuscript that addresses the points raised during the review process.

The reviewers raised some important issues that need to be fully addressed, namely: 

1 - According to PLOS ONE guidelines: “if the manuscript’s primary purpose is the description of new software or a new software package, this software must be open source, deposited in an appropriate archive, and conform to the Open Source Definition.”

2 - Give more details about the methods used so others can replicate the analysis, include error analysis and examples

3 - Include in the analysis the BioASQ 16-18 datasets and systems.

We would appreciate receiving your revised manuscript by Feb 20 2020 11:59PM. To enhance the reproducibility of your results, we recommend that if applicable you deposit your laboratory protocols in protocols.io, where a protocol can be assigned its own identifier (DOI) such that it can be cited independently in the future. For instructions see: http://journals.plos.org/plosone/s/submission-guidelines#loc-laboratory-protocols

We look forward to receiving your revised manuscript.

Kind regards,

Francisco M Couto

Academic Editor

PLOS ONE

Journal Requirements:

We note that one or more of the authors are employed by a commercial company: Alibaba Group.

Reviewers' comments:

Reviewer's Responses to Questions

**Comments to the Author**

1. Is the manuscript technically sound, and do the data support the conclusions?

Reviewer #1: No

Reviewer #2: Yes

Reviewer #3: Yes

2. Has the statistical analysis been performed appropriately and rigorously? 

Reviewer #1: Yes

Reviewer #2: Yes

Reviewer #3: Yes

3. Have the authors made all data underlying the findings in their manuscript fully available?

Reviewer #1: Yes

Reviewer #2: Yes

Reviewer #3: Yes

4. Is the manuscript presented in an intelligible fashion and written in standard English?

Reviewer #1: No

Reviewer #2: Yes

Reviewer #3: Yes

5. Review Comments to the Author

Reviewer #1: The manuscript provides an approach to snippet retrieval for biomedical question answering using recursive auto-encoders (RAE). The authors evaluated their approach on BioASQ datasets and compared to other deep learning architectures such as CNN, RNN and LSTMs, as well as with BioASQ participants. However it's conclusions are not well supported since the comparisons are not made against the current state-of-the-art. Still, I believe that if the appropriate comparisons are made (i.e. evaluate on the most recent BioASQ datasets and compare with the results obtained by the more recent systems) and the results are discussed with more detail, the idea is interesting and has potential.

strengths:

- statistical comparison with other approaches

- evaluation on multiple BioASQ datasets

- detailed explanation of the proposed method with figures

weaknesses:

- The results are not well supported. To really understand why your method outperforms all the others, I would need more than MAP scores, for example, questions that other method got wrong and your method got right. Other aspects such as inference time would also be interesting.

- The discussion does not go in-depth about what makes your method "more suitable than other vector representation models to a large extent" or that it "retains as much of the local semantics as possible".

- Why not show the results for BioASQ 16 -18? You also claim that your approach was evaluated on two other open domain tasks on the abstract, but this is never mentioned in the text.

- Furthermore you also ignored other approaches to the snippet ranking task that have been proposed since the original tasks. For example this paper: https://www.sciencedirect.com/science/article/pii/S1532046417300503 (Table 2) shows esults for BioASQ 2015 snippet retrieval task superior to yours on every batch.

- Since the code is not provided, it would be difficult to test on other datasets or reproduce the results

- Please make clear the distinction between Recurrent Neural networks and Recursive Neural Networks. You end up using RNN for both, while generally RNN is used for Recurrent Neural Networks.

- There are occasional typos and language errors that should be revised, for example Line 250 "re relevant", while some expressions do not seem to be standard English: Line 147 "From our points".

Reviewer #2: This paper presents a model for biomedical question answering using a neural network architecture. The proposed model uses a recursive autoencoder aggregated over a binary tree representation of the sequence and a recurrent neural net is trained with a ranking loss of candidate word spans. Results are reported on three years of BioASQ challenges comparing against different encoding methods and competing systems at the challenge.

Strengths

- The paper presents an extensive evaluation on the BioASQ challenge

- The results outperform baseline systems significantly. This is presented through statistical testing and comparison with multiple baselines

- Baselines range from simple IR systems to other system implementations in the challenge

Weaknesses

- There is no error analysis documenting the reasons why the proposed system performs better

- It is unclear if the code will be publicly available

- It is unclear how the system will perform outside the biomedical domain and on datasets other than the BioASQ task

- The paper needs some proof reading for English

- Why is a binary tree better than a parse tree for the RAE?

Overall I think this is good work but needs to convince the reader better the reasons behind why the proposed model outperforms existing methods.

Reviewer #3: This paper addresses the problem of extracting and ranking the

relevant snippets in the context of Question-Answering. To achieve

this goal, the authors proposes to catch the semantic relation between

the question and the snippet and introduce a ranking error.

The paper is well written and the method is clearly described.

The state of the art is well covered. The authors may also mention

work related to the querying biomedical linked data, even if NLP is

mainly involved in the question processing.

The method is evaluated against standard QA set from BioASQ

challenges. A significance testing is performed However, the results

could be better analysed, especially to identify if the improvement

are related a better ranking or a better snippet extraction.

A discussion about the limit of the message is also welcomed.

There is a regular typo along the article: many times, space character

is wrongly added before commas or periods (even in the affliation part).

line 250: re -> are

6. PLOS authors have the option to publish the peer review history of their article (what does this mean?). If published, this will include your full peer review and any attached files.

Reviewer #1: Yes: Andre Lamurias

Reviewer #2: No

Reviewer #3: No

---

## [Author Response · Author response to Decision Letter 0]

20 Feb 2020

The reviewer's comments have been answered item by item in the attachment, and the response has been uploaded as a separate file.

---

## [Decision Letter · Decision Letter 1]

19 May 2020

PONE-D-19-28201R1

List-wise Learning to Rank Biomedical Question-Answer Pairs with Deep Ranking Recursive Autoencoders

PLOS ONE

Dear Miss. Yan,

Thank you for submitting your manuscript to PLOS ONE. After careful consideration, we feel that it has merit but does not fully meet PLOS ONE’s publication criteria as it currently stands. Therefore, we invite you to submit a revised version of the manuscript that addresses the points raised during the review process.

There are still important issues raised by the reviewers that need to be addressed, namely inconsistencies in the MAP scores, lack of examples/error analysis, incomplete comparison of BioASQ2015, and grammar errors.

We would appreciate receiving your revised manuscript by Jul 03 2020 11:59PM. To enhance the reproducibility of your results, we recommend that if applicable you deposit your laboratory protocols in protocols.io, where a protocol can be assigned its own identifier (DOI) such that it can be cited independently in the future. For instructions see: http://journals.plos.org/plosone/s/submission-guidelines#loc-laboratory-protocols

We look forward to receiving your revised manuscript.

Kind regards,

Francisco M Couto

Academic Editor

PLOS ONE

Reviewers' comments:

Reviewer's Responses to Questions

**Comments to the Author**

1. If the authors have adequately addressed your comments raised in a previous round of review and you feel that this manuscript is now acceptable for publication, you may indicate that here to bypass the “Comments to the Author” section, enter your conflict of interest statement in the “Confidential to Editor” section, and submit your "Accept" recommendation.

Reviewer #1: (No Response)

Reviewer #4: (No Response)

2. Is the manuscript technically sound, and do the data support the conclusions?

Reviewer #1: No

Reviewer #4: Partly

3. Has the statistical analysis been performed appropriately and rigorously? 

Reviewer #1: Yes

Reviewer #4: I Don't Know

4. Have the authors made all data underlying the findings in their manuscript fully available?

Reviewer #1: Yes

Reviewer #4: Yes

5. Is the manuscript presented in an intelligible fashion and written in standard English?

Reviewer #1: No

Reviewer #4: No

6. Review Comments to the Author

Reviewer #1: The authors addressed some of my issues properly, namely they made the code open source, explained with more the detail the advantages of their method, provided results of more recent edition of BioASQ and mention post-competiton results of other groups.

However some issues were not properly addressed and others were introduced with this revision:

- examples/error analysis: unlike the teams that participated on the competition, it is possible to analyse the test set and give examples where the system performed well and where it failed. This is helpful to improve future approaches but it is still not present in this version

- proper comparison to other methods: although the authors now show the results of their method on more recent editions of BioASQ; they justify the lower performance of the proposed method on the recent editions this way: "The decreases of improvements do not indicate the decline of robustness. The ultimate causes are the introduction of extra resources. Especially after BioASQ 2015, most of the systems based on extra resources contain a large amount of domain knowledge in biomedicine." this statement is vague and unfounded: what do the authors mean by "extra resources"? The description provided on the manuscript of the other team's systems mentions other resources used on the previous editions too: "Hana Database and BioPortal to retrieve biomedical concepts" and "look up the synonyms of the keywords in queries to select effective terms for query expansion". A more in-depth analysis is necessary to understand if the extra resources were really the cause of the improved performance of recent systems.

- Furthermore although it is mentioned in the related work section, the results of the proposed method on BioASQ2015 are not compared with these: https://www.sciencedirect.com/science/article/pii/S1532046417300503

- English should be revised, particularly on the text that was added since the previous version

-Tables 5 6 and 7 - it is misleading to have the values of the first row bolded. I would suggest highlighting the highest value of each column instead.

Reviewer #4: This paper presents an approach for the automated selection of relevant snippets for answering biomedical questions. The authors propose a Deep Learning approach based on ranking question-snippet pairs introducing a “ranking error” loss function for this task. Different experiments are presented comparing the performance of the proposed approach to various baselines based on classic IR models (implemented by them) and to some existing systems participating in the BioASQ challenge. Their system outperforms the IR baselines and achieves decent results compared with the BioASQ participants.

I consider two main improvements necessary for this manuscript:

1) There are inconsistencies in the MAP scores of BioASQ participants reported in tables 2 to 7. A thorough check is needed to guarantee that the numbers are correct. 1.a) For example, the MAP scores in table 4 are significantly lower than the official results http://participants-area.bioasq.org/results/3b/phaseA/. This table even includes a score in the second batch for the oaqa system which didn't participate in that batch. In table 5, batch 5 the KNU team hast the score of the ustb team and vice versa. 1.b) It is not clear which teams are selected to be presented in the tables. For example in table 4 (batch 4 and 5) and 5 (batch 2,3,4, and 5) the winning fdu team (with the highest MAP score) is not included in the table. The same with the aueb team in table 7. 1.c) In addition, even for teams included in the tables, the reported MAP score is not always the best score achieved by the team. For example, in table 6, batch 1, the score 0.1620 is reported which corresponds to utsb_prir3, while utsb_prir2 achieves 0.1774.

2) There are various expression and orthographic issues in the manuscript that need to be checked. Some examples:

line 46: improvements of performances, -> improvements of performance,

line 88: So the extraction of documents is a great challenge -> So the extraction of snippets is a great challenge

line 102: Like the BioASQ challenge, participants of the BioNLP-> Similarly to participants of the BioASQ challenge, participants of the BioNLP...

line 122: Sarrouti etc. [34] proposed using temmed words -> Sarrouti and El Alaoui [34] proposed the use of stemmed words

line 153: From our perspectives, -> From our perspective,

line 182: on evey input questions and remove all non-noun phrase(NNP) part-> on every input question and remove all non-noun phrase (NNP) parts

line 186: oftenly. -> often.

line 187: since search engine tend to retrieve -> since search engines tend to retrieve

line 189: works best then leave nouns only -> works better than leaving nouns only

line 192: the semantic vectors of words are requested -> the semantic vectors of words are required

line 194: Medline Articles collection -> Medline article collection

line 206: standards which and we can -> (need for rephrasing)

line 208: more suitably and easier to encode vectors together -> (need for rephrasing)

line 251: distribution of with and without -> distribution with and without

line 309: "What -> ``What

line 313: Comparison Algorithms -> Algorithm Comparison (or Comparison of Algorithms)

line 350: after a series of compute -> (need for rephrasing)

line 352: The reason why the vector y can returned to the input vectors is the vector y has most -> The reason why the vector y can be returned to the input vectors is that the vector y has most

Fig 4. Input vector x1; x2 is the children mentioned in Recursive Autoencoders and Variants section, and encoded vector y is the parent mentioned in Recursive Autoencoders and Variants section. -> Fig 4. The input vectors x1; x2 are the children mentioned in Recursive Autoencoders and Variants section, and the encoded vector y is the parent mentioned in Recursive Autoencoders and Variants section.

line 377: other teams and our exact MAP scores -> (need for rephrasing)

line 456: The reason why they get a better result mainly because -> They get a better result mainly because

7. PLOS authors have the option to publish the peer review history of their article (what does this mean?). If published, this will include your full peer review and any attached files.

Reviewer #1: Yes: Andre Lamurias

Reviewer #4: No

---

## [Author Response · Author response to Decision Letter 1]

6 Jul 2020

Thank you for your responsible attitude and sincere suggestions. All comments were carefully answered in the ”response to reviewer". Thanks again.

---

## [Decision Letter · Decision Letter 2]

12 Oct 2020

PONE-D-19-28201R2

List-wise Learning to Rank Biomedical Question-Answer Pairs with Deep Ranking Recursive Autoencoders

PLOS ONE

Dear Dr. Yan,

Thank you for submitting your manuscript to PLOS ONE. After careful consideration, we feel that it has merit but does not fully meet PLOS ONE’s publication criteria as it currently stands. Therefore, we invite you to submit a revised version of the manuscript that addresses the points raised during the review process.

The manuscript was recommended to minor revision, thus please make changes according to the suggested comments and please do a final proofreading of the document. Optionally, if you find relevant, you can update your results according to a recent corpus that also improved BioASQ IR tasks: https://ieeexplore.ieee.org/document/9184044

We look forward to receiving your revised manuscript.

Kind regards,

Francisco M Couto

Academic Editor

PLOS ONE

Reviewers' comments:

Reviewer's Responses to Questions

**Comments to the Author**

1. If the authors have adequately addressed your comments raised in a previous round of review and you feel that this manuscript is now acceptable for publication, you may indicate that here to bypass the “Comments to the Author” section, enter your conflict of interest statement in the “Confidential to Editor” section, and submit your "Accept" recommendation.

Reviewer #1: All comments have been addressed

Reviewer #4: (No Response)

2. Is the manuscript technically sound, and do the data support the conclusions?

Reviewer #1: Yes

Reviewer #4: Yes

3. Has the statistical analysis been performed appropriately and rigorously? 

Reviewer #1: Yes

Reviewer #4: I Don't Know

4. Have the authors made all data underlying the findings in their manuscript fully available?

Reviewer #1: Yes

Reviewer #4: Yes

5. Is the manuscript presented in an intelligible fashion and written in standard English?

Reviewer #1: Yes

Reviewer #4: Yes

6. Review Comments to the Author

Reviewer #1: (No Response)

Reviewer #4: This paper presents an approach for the automated selection of relevant snippets for answering biomedical questions. As in the original version of the manuscript, the authors present experiments comparing their deep-learning approach that introduces a "ranking error" loss function for ranking question-snippet pairs.

Considerable improvement has been done in the direction of the two issues highlighted previously. However, there are still some minor improvements that need the attention and action of the authors.

1) The main inconsistencies in the MAP scores of BioASQ participants presented in tables 2-7 have been removed. However, a) there is an error in table 6, batch 3 as the results reported are for a different task (document retrieval) instead of snippet retrieval, as done for the other batches and tables. b) The performance of the top participating systems is still missing in some tables/batches (In particular: table 3, batch 2, HPI-S1 MAP 0.048; table 4, batch 1, HPI-S2, MAP 0.0971; table 4, batch 3, oaqa, MAP 0.1969; table 4, batch 4, fdu2, MAP 0.2480; table 4, batch 5, fdu2, MAP 0.2424; table 5, batch 1, HPI-S2, MAP 0.1601)

2) There are still some syntactic issues in English that should be carefully re-checked.

In addition, a minor formatting error exists in Table 1, where the performance 0.1203 is bold instead of 0.1412 in BioASQ 2015, batch 5.

7. PLOS authors have the option to publish the peer review history of their article (what does this mean?). If published, this will include your full peer review and any attached files.

Reviewer #1: No

Reviewer #4: No

---

## [Author Response · Author response to Decision Letter 2]

24 Oct 2020

We are very grateful for your comments and opinions , and apologize for our mistakes.

---

## [Editor Report · Decision Letter 3]

27 Oct 2020

List-wise Learning to Rank Biomedical Question-Answer Pairs with Deep Ranking Recursive Autoencoders

PONE-D-19-28201R3

Dear Dr. Yan,

We’re pleased to inform you that your manuscript has been judged scientifically suitable for publication and will be formally accepted for publication once it meets all outstanding technical requirements.

Kind regards,

Francisco M Couto

Academic Editor

PLOS ONE
---

## [Editor Report · Acceptance letter]

29 Oct 2020

PONE-D-19-28201R3 

List-wise Learning to Rank Biomedical Question-Answer Pairs with Deep Ranking Recursive Autoencoders 

Dear Dr. Yan:

I'm pleased to inform you that your manuscript has been deemed suitable for publication in PLOS ONE. Congratulations! Your manuscript is now with our production department. 

Kind regards, 

on behalf of

Mr. Francisco M Couto 

Academic Editor

PLOS ONE